# LEARNING STRUCTURED SPARSE NEURAL NETWORKS USING GROUP ENVELOPE REGULARIZATION

## ABSTRACT

We propose an efficient method to learn both unstructured and structured sparse neural networks during training, utilizing a novel generalization of the sparse envelope function (SEF) used as a regularizer, termed *weighted group sparse envelope function* (WGSEF). The WGSEF acts as a neuron group selector, which is leveraged to induce structured sparsity. The method ensures a hardware-friendly structured sparsity of a deep neural network (DNN) to efficiently accelerate the DNN's evaluation. Notably, the method is adaptable, letting any hardware specify group definitions, such as filters, channels, filter shapes, layer depths, a single parameter (unstructured), etc. Owing to the WGSEF's properties, the proposed method allows to a pre-define sparsity level that would be achieved at the training convergence, while maintaining negligible network accuracy degradation or even improvement in the case of redundant parameters. We introduce an efficient technique to calculate the exact value of the WGSEF along with its proximal operator in a worst-case complexity of $O(n)$, where $n$ is the total number of group variables. In addition, we propose a proximal-gradient-based optimization method to train the model, that is, the non-convex minimization of the sum of the neural network loss and the WGSEF. Finally, we conduct an experiment and illustrate the efficiency of our proposed technique in terms of the completion ratio, accuracy, and inference latency.

## 1 INTRODUCTION

In the last decade, far-reaching progress has been achieved in the study of neural networks, as these achieve the best performance in many machine learning tasks. Nonetheless, modern networks are becoming gradually larger with an enormous number of parameters with the cost of rising storage, memory footprint, computing resources and energy consumption Deng et al. (2020); Cheng et al. (2017). Notwithstanding, it has been shown in recent studies Han et al. (2015a); Ullrich et al. (2017); Molchanov et al. (2017) that the latest neural networks tend to be highly over-parametrized in the sense that a relatively large amount of parameters are redundant and could have been pruned (or, nullified) without degrading the network precision.

Over-parametrization has two notable problematic effects: first, it can very easily results in overfitting Allen-Zhu et al. (2019), as well as memorizing random patterns in the data Zhang et al. (2021), leading to inferior generalization. Second, it becomes quite challenging to deploy these networks to low hardware edge devices due to, e.g., high computational cost results in a large consumption of power and a long inference time, and a substantial redundant memory space for storing is needed (which obviously would never be available on edge devices). This has led to the search for sparse efficient and effective neural network architectures which has become increasingly a front challenge.

The most common way to tackle the first issue, namely, avoiding overfitting for better generalization is, to use properly suitable regularization, while the accepted approach to address the second problem is to apply post-training model compression techniques. However, the sparsification technique, i.e., regularizing the number of parameters during training, would result in both model compression and overfit-regularization.

Quite recently, the idea of structured sparsification was used in Wen et al. (2016b); Bui et al. (2021) to learn sparse neural networks that leverage tensor arithmetic in dedicated neural processing units

(NPUs). In a nutshell, structured sparsity learning amounts to inducing sparsity onto structured components (e.g., channels, filters, or layers) in the neural network during the optimization procedure. This leads, in practice, to both low latency and lower power consumption, which can not be obtained by deploying unstructured sparse models on such modern hardware.

In the case of the unstructured sparsity-inducing, the most natural regularizer would be the so-called $\ell_0$-pseudo-norm function that counts the number of nonzero elements in the input vector, i.e., $\|\mathbf{z}\|_0 \triangleq |\{i : z_i \neq 0\}|$. These sparse regularized minimization/ training problems are of the form $\min_{\mathbf{z} \in \mathbb{R}^n} \{f(\mathbf{z}) + \lambda \|\mathbf{z}\|_0\}$, or, alternatively, one can explicitly constrain the number of parameters used for regression and solve $\min_{\mathbf{z} \in \mathbb{R}^n} \{f(\mathbf{z}) : \|\mathbf{z}\|_0 \leq k\}$. Unfortunately, the $\ell_0$-norm is a difficult function to handle being non-convex and even non-continuous. Indeed, these types of regression models are known to be NP-hard problems in general Natarajan (1995). The only known algorithm guaranteed to output the global optimal solution is essentially a brute-force search over all possible subsets components of the vector $\mathbf{z}$. In practice, due to this expensive computational cost, the global optimal solution can not be computed in a reasonable time, even for a very small number of parameters, precluding its use in large neural networks.

As a remedy for the inherent problem above, Beck & Refael (2022) proposed a highly efficient tractable tight convex relaxation technique, termed sparse envelope function (SEF), for the sum of both $\ell_0$ and $\ell_2$ norms. Specifically, Beck & Refael (2022) suggested using this relaxation as a regularizer term for a convex loss objective, particularly for a linear regression model, to achieve feature selection while explicitly limiting the number of features to be a fixed parameter $k$. It was also shown that the performance of the regularization method in both reconstruction of a sparse noisy signal and recovering its support, surpass the performance of state-of-the-art techniques, such as, the Elastic-net Zou & Hastie (2005), $k$-support norm Argyriou et al. (2012), etc. Also, it was shown that the computational complexity of the SEF approach is linear in the number of parameters, while all others requires at least quadratic in the number of features, and thus very attractive.

With the goal of enabling structured sparsification learning that can be customized for different NPU devices, in this papaer we propose a novel generalized notion of the SEF regularizer to handle group structured sparsification in neural network training. Our new generalized regularization term selects the most essential $k \leq m$ predefined groups of neurons (which could be convolutional filters, channels, individual neurons, or any other user-defined/NPU definition) and prunes all others, while maintaining minimal network accuracy degradation. We define the new regularization term mathematically, propose an efficient method to calculate its value and proximal operator, and suggest a new algorithm to solve the complete optimization problem involving the non-convex term, which is the composition of the loss function and the neural network output.

**Related work.** The topic of regularization-based pruning received a lot of attention in recent years. Generally speaking, these studies can be divided into unstructured and structured pruning. Most prominent regularizers are the convex $\ell_1$ and $\ell_2$ norms Liu et al. (2017); Ye et al. (2018); Han et al. (2015a;b), as well as the non-convex $\ell_0$ "norm" Louizos et al. (2017); Han et al. (2015c), where Bayesian methods and additional regularization terms for practicality, were used to deal with the non-convexity of the $\ell_0$ norm. Additional works of Donoho & Elad (2003); com (2021-2023) suggest methods for norm $l_0$ relaxation by employing $l_1$ minimization in general (nonorthogonal) dictionaries and leading to an error surface with fewer local minima than the $l_0$ norm. The motivation for these regularizers is their "sparsity-inducing" property which can be harnessed to learn sparse neural networks. While these fundamental papers significantly reduce the storage needed to store the networks on hardware, There were no benefits in reducing the inference latency time or either in cutting down power consumption. That is, the sparse neural networks, learned by the aforementioned methods, were not adapted to the structure of the hardware they aimed to run on. This observation led researchers to propose regularization-based structured pruning in favor of accelerating the running time. For example, Lebedev & Lempitsky (2015); Wen et al. (2016a); Yuan & Lin (2006) proposed the use of the Group Lasso regularization technique to learn sparse structures, and Scardapane et al. (2017) uses Sparse Group Lasso, summing Group Lasso with the standard Lasso penalty. Other convex regularizers include the Combined Group and Exclusive Sparsity (CGES) Yoon & Hwang (2017), which extends Exclusive Lasso (in essence, squared $\ell_1$ over groups) Zhou et al. (2010) using Group Lasso. Recently, Bui et al. (2021) suggested a family of nonconvex regularizers that blend Group Lasso with nonconvex terms ($\ell_0$, $\ell_1 - \ell_2$ Lou et al. (2015), and SCAD Fan & Li (2001)). Since Bui et al. (2021) introduces non-convexity term into the penalty, it also requires

an appropriate optimization scheme, for which the authors propose an Augmented Lagrangian type method. However, this optimization algorithm has an inner optimization loop with a high computational cost. Moreover, their extensive experiments do not show an accuracy or sparsity advantages over convex penalties, suggesting that it might be still desirable to use a convex regularizer. Other methods, such as Chen et al. (2021); Li et al. (2019), focus on a group structure that captures the relations between parameters, neurons, and layers, in order to construct groups that can maximize network compression while minimizing accuracy loss. However, these methods still apply Group Lasso regularization. Specifically in Chen et al. (2021), the authors introduce the concept of Zero-Invariant Groups (ZIGs), which includes all input and output connections between layers. In the context of CNNs, it extends the channel-wise grouping Wen et al. (2016b) to include corresponding batch normalization parameters. By using this group structure, entire blocks of parameters can be removed while keeping dimensions aligned between layers, and ultimately allowing network compression. Moreover, their optimization scheme utilizes a two-phase algorithm to include a half-space projection step, which they name HSPG.

Finally, we mention that there exist other techniques for neural net compression, such as, quantization, low-rank decomposition, to name a few. In quantization, Courbariaux et al. (2016); Rastegari et al. (2016); Gong et al. (2014), the precision of the weights is reduced, by representing weights using a low number of bits (i.e., 8-bit) instead of higher one (i.e., 32-bit floating point values). The low-rank decomposition approach Denton et al. (2014); Jaderberg et al. (2014); Lebedev et al. (2014) is based on the observation that many weight matrices in neural networks are highly correlated and can be well approximated by matrices with a lower rank. By decomposing a weight matrix into lower-rank matrices, one can reduce the total number of parameters in the network.

**Notation.** We denote $\mathbf{e}$ for the vector of all ones. For a positive integer $m$, we denote $[m] \equiv \{1, 2, \ldots, m\}$. Let $f : \mathbb{R}^n \to \mathbb{R}$, be an extended real-valued function. Then, the conjugate function of $f$, denoted by $f^\star : \mathbb{R}^n \to \mathbb{R}$ is defined as $f^\star(y) = \max_{x \in \mathbb{R}^n} \{\langle x, y \rangle - f(x)\}$, for any $y \in \mathbb{R}^n$. The bi-conjugate function is defined as the conjugate of the conjugate function, i.e., $f^{\star\star}(x) = \max_{y \in \mathbb{R}^n} \{\langle x, y \rangle - f^\star(y)\}$, for any $x \in \mathbb{R}^n$. Finally, the proximal operator of a proper, lower semi-continuous convex function $f : \mathbb{R}^n \to \mathbb{R}$ is defined as $\operatorname{prox}_f(v) = \arg\min_{x \in \mathbb{R}} \{f(x) + \frac{1}{2}\|x - v\|_2^2\}$, for any $v \in \mathbb{R}^n$.

## 2 PROBLEM FORMULATION

In this section, we formulate the problem and introduce the weighted group sparse envelop function. Without loss of generality, our method is formulated on weights sparsity, but it can be directly extended to neuron sparsity (i.e., both weights and bias). Let $\mathcal{D}$ be a dataset consisting of $N$ i.i.d. input output pairs $\{(x_1, y_1), \ldots, (x_N, y_N)\}$. The neural network training problem is formalized as the following empirical risk minimization procedure on the parameters $\theta \in \Theta$ of a hypothesis $f(\cdot; \theta)$,

$$\operatorname*{argmin}_{\theta \in \Theta} \frac{1}{N} \sum_{i=1}^{N} \mathcal{L}(f(x_i, \theta), y_i), \tag{1}$$

where $\mathcal{L}(\cdot)$ corresponds to a loss function, e.g., cross-entropy loss for classification, mean-squared error for regression, etc. Below, $n \triangleq |\theta|$ denotes the number of parameters. The training of neural networks is prone to overfitting, therefore, regularization is typically employed. Consider a regularized empirical risk minimization procedure with a regularizer $\Omega(\cdot) : \Theta \to \mathbb{R}$ on the parameters $\theta$ of a hypothesis $f(\cdot; \theta)$,

$$\operatorname*{argmin}_{\theta \in \Theta} \frac{1}{N} \sum_{i=1}^{N} \mathcal{L}(f(x_i, \theta), y_i) + \lambda \cdot \Omega(\theta), \tag{2}$$

where $\lambda \in \mathbb{R}$. Weight-decay, known as the $\ell_2$-norm regularization, is the most common technique for deep network training regularization (in this case, $\Omega(\theta) = \frac{1}{2}\|\theta\|_2^2$). It prevents overfitting and improves generalization as it enforces the weights to decrease proportionally to their magnitudes.

The most natural way to force a predefined $k$-level sparsity would be to constraint the number of non-zeros parameters (e.g., the model weights), which can be done by adding the constraint that

$\|\theta\|_0 \leq k$, where $k \leq n$ is the required predefined level of sparsity. In this case, the training problem is formalized as follows,

$$\operatorname*{argmin}_{\theta \in \Theta} \frac{1}{N} \sum_{i=1}^{N} \mathcal{L}\left(f\left(x_i, \theta\right), y_i\right) + \frac{\lambda}{2}\|\theta\|_2^2 \quad \text{s.t.} \quad \|\theta\|_0 \leq k. \tag{3}$$

We refer to the above training problem as *unstructured sparsity*. In the case of *structured sparsity*, the parameters $\theta$ are divided into *predefined* disjoint sub-groups. These subgroups could define building blocks architecture of DNN's, i.e., filters, channels, filter shapes, and layer depth. Consider the following definition.

**Definition 1** (Projection). *Let $s$ be a subset of indexes $s \subseteq \{1, 2, \ldots, n\}$ of size $|s| \leq n$. Then, given some vector $\theta \in \mathbb{R}^n$, the projection $M_s : \mathbb{R}^n \to \mathbb{R}^n$ preserves only the entries of $\theta$ that belong to the set $s$. Furthermore, let $A_s$ be an $n \times n$ diagonal matrix, where $[A_s]_{ii} = 1$ if $i \in s$, and zero, otherwise. Note that $M_s(\theta) = A_s \theta$.*

**Example 1.** *Let $n = 3, \theta = (3, 6, 9)^\top, s = \{1, 3\} \subseteq [n]$, and accordingly $|s| = 2$, then $M_s(\theta) = M_s\left((3, 6, 9)^\top\right) = (3, 0, 9)^\top$, with $[A_s]_{11} = [A_s]_{33} = 1$, and zero otherwise.*

Following the above definition, let $m \leq n$ subsets $s_1, s_2, \ldots, s_m$ be a given partition of $[n]$, namely, $s_i \cap s_j = \emptyset$, for all $i \neq j$, and $\bigcup_{i=1}^m s_i = [n]$. Without loss of generality, we assume that $n\%m = 0$; otherwise, the groups would have different coordinates. Every group is associated with some weight $d_j \in \mathbf{R}_{++}$, where $j \in [m]$, e.g., $d_j = \frac{1}{|s_j|}$, namely, we normalize by the group size. For simplicity of notation, let $\theta s_i = M_{s_i}(\theta)$, for $i = 1, 2, \ldots, m$. Then, our structured training problem is,

$$\operatorname*{argmin}_{\theta \in \Theta} \frac{1}{N} \sum_{i=1}^{N} \mathcal{L}\left(f\left(x_i, \theta\right), y_i\right) + \frac{\lambda}{2} \sum_{j=1}^{m} d_j \|\theta s_j\|_2^2 \tag{4}$$

$$\text{s.t.} \quad \left\|\|\theta s_1\|_2^2, \|\theta s_2\|_2^2, \ldots, \|\theta s_m\|_2^2\right\|_0 \leq k.$$

To wit, we constrain the number of groups which has at least one non-zero coordinate, to be at most $k$. Let $C_k$ denote the set of all $k$ sparse groups, i.e.,

$$C_k \triangleq \left\{\theta : \left\|\|\theta s_1\|^2, \|\theta s_2\|^2 \ldots, \|\theta s_m\|^2\right\|_0 \leq k\right\}, \tag{5}$$

and define $\delta_{C_k}$ as the following extended real-valued function,

$$\delta_{C_k}(\Theta) \triangleq \begin{cases} 0, & \left\|\|\Theta s_1\|^2, \|\Theta s_2\|^2 \ldots, \|\Theta s_m\|^2\right\|_0 \leq k, \\ \infty, & \text{else.} \end{cases} \tag{6}$$

Then, the optimization problem in equation 4 can be reformulated as,

$$\operatorname*{argmin}_{\theta \in \Theta} \frac{1}{N} \sum_{i=1}^{N} \mathcal{L}\left(f\left(x_i, \theta\right), y_i\right) + \lambda \cdot gs_k(\theta), \tag{7}$$

where $gs_k(\theta) \triangleq \frac{1}{2} \sum_{j=1}^m d_j \|\theta s_j\|_2^2 + \delta_{C_k}(\Theta)$. Equivalently, $gs_k(\theta)$ can be rewritten as $gs_k(\theta) \triangleq \frac{1}{2} \sum_{i=1}^n d_i \cdot \theta_i^2 + \delta_{C_k}(\Theta)$, where $d_i = d_j$ for every $i \in s_j$.

The $\ell_0$-norm that appears in 7, is a difficult function to handle being nonconvex and even non-continuous, making the problem an untraceable combinatorial NP hard problem Natarajan (1995). Following Beck & Refael (2022), one approach to deal with this inherent difficulty is to consider the best convex estimator of $gs_k(\cdot)$. The later is its biconjgate function, namely, $\mathcal{GS}_k(\theta) = gs_k^{\star\star}(\theta)$, which we refer to as the *weighted group sparse envelope function (WGSEF)*.

**Remark 1.** *Consider the case $m = n$, namely, every subset $s_i, i \in [m]$ is a singleton and $\forall j \in [m], d_j = 1$. Here, $gs_k(\theta) = s_k(\theta)$, where $s_k(\theta) = \frac{1}{2}\|\theta\|_2^2$ if $\|\theta\|_0 \leq k$ and $s_k(\theta) = \infty$, otherwise. Accordingly, in this case, $\mathcal{GS}_k(\theta) = \mathcal{S}_k(\theta) = s_k^{\star\star}(\theta)$, and $s_k^{\star\star}(\theta)$ is exactly the classical SEF, namely, $\mathcal{S}_k(\cdot)$. Therefore, $gs_k(\cdot)$ is indeed a new generalization of SEF to handle group sparsity.*

Thus, the path taken in this paper is to consider the following relaxation training/learning problem,

$$\operatorname*{argmin}_{\theta \in \Theta} \frac{1}{N} \sum_{i=1}^{N} \mathcal{L}\left(f\left(x_i, \theta\right), y_i\right) + \mathcal{GS}_k(\theta). \tag{8}$$

In the following section, we develop an efficient algorithm to calculate the value and the prox-operator Beck (2017) of WGSEF; these will play an essential ingredient when solving (8).

## 3 TIGHT CONVEX RELAXATION

Let us start by introducing some notation. For any $\theta \in \mathbb{R}^n$ and $m$ subgroups of indexes $s_1, s_2, \ldots, s_m \in [n]$, we denote by $M_{\langle s_i \rangle}(\theta)$ the corresponding subgroup of coordinates in $\theta$ with the $i$th largest $\ell_2$-norm, i.e.,

$$\left\| M_{\langle s_1 \rangle}(\theta) \right\|_2 \geq \left\| M_{\langle s_2 \rangle}(\theta) \right\|_2 \geq \ldots \geq \left\| M_{\langle s_m \rangle}(\theta) \right\|_2, \ \forall i \neq j \in [m].$$

We next show that the conjugate of the $k$ group sparse envelopes is the $k$ weighted group hard thresholding function. In the sequel, we let $D$ be the $n \times n$ diagonal positive-definite weights matrix, such that $D_{i,i} = \sqrt{d_i}, \forall i \in s_j$, and $j \in [m]$.

**Lemma 3.1** (The $k$ weighted group sparse envelop conjugate). *Let subsets $s_1, s_2, \ldots, s_m$ be a set of $m \leq n$ disjoint indexes that partition $[n]$. Then, for any $\tilde{\theta} \in \mathbb{R}^n$,*

$$gs_k^{\star}(\tilde{\theta}) = \frac{1}{2} \sum_{j=1}^{k} \frac{1}{d_j} \left\| M_{\langle s_j \rangle}(\tilde{\theta}) \right\|_2^2. \tag{9}$$

Next, we obtain the bi-conjugate function of the $k$ weighted group sparse envelope.

**Lemma 3.2** (The variational bi-conjugate $k$ weighted group sparse envelop ). *Let $s_1, s_2, \ldots, s_m$ be a set of $m \leq n$ disjoint subsets that partition $[n]$. Then, for any $\theta \in \mathbb{R}^n$, the bi-conjugate of the $k$ group sparse envelop is given by*

$$\mathcal{GS}_k(\theta) = \frac{1}{2} \min_{\mathbf{u} \in B_k} \left\{ \sum_{j=1}^{m} d_j \phi \left( A_{s_j} \theta, u_j \right) \right\}, \tag{10}$$

*where*

$$\phi \left( A_{s_j} \theta, u_j \right) \triangleq \begin{cases} \frac{\theta^\top A_{s_j} \theta}{u_j}, & u_j > 0, \\ 0, & u_j = 0 \cap A_{s_j} \theta = 0, \\ \infty & \text{else.} \end{cases} \tag{11}$$

The following is a straightforward corollary of Lemma 3.2.

**Corollary 3.2.1.** *The following holds:*

$$\mathcal{GS}_k(\theta) = \mathcal{S}((\sqrt{d_1} \| A_{s_1} \theta \|_2, \sqrt{d_2} \| A_{s_2} \theta \|_2, \ldots, \sqrt{d_m} \| A_{s_m} \theta \|_2)^\top), \tag{12}$$

*where $\mathcal{S}(\theta) \triangleq s_k^{\star\star}(\theta)$ is the standard SEF.*

The implication of the above corollary is that in order to calculate $\mathcal{GS}_k(\theta)$ we only need to apply an algorithm that calculates the SEF at $(\sqrt{d_1} \| A_{s_1} \theta \|_2, \sqrt{d_2} \| A_{s_2} \theta \|_2, \ldots, \sqrt{d_m} \| A_{s_m} \theta \|_2)^\top$.

**Remark 2.** *Noting $\| \sqrt{d_j} A_{s_j} \theta \|_2^2 = \sum_{i \in s_j} d_j \theta_i^2$ we observe that since $s_j$ is given, the number of operations required to calculate $\| \sqrt{d_j} A_{s_j} \theta \|_2^2$ is linear w.r.t. $|s_j|$. Thus, the computational complexity of calculating the vector $(\sqrt{d_1} \| A_{s_1} \theta \|_2, \sqrt{d_2} \| A_{s_2} \theta \|_2, \ldots, \sqrt{d_m} \| A_{s_m} \theta \|_2)^\top$ is linear in $n$.*

## 4 PROXIMAL MAPPING OF THE WGSEF

In this section, we will show how to efficiently compute the proximal operator of positive scalar multiples of $\mathcal{GS}_k$. The ability to perform such an operation implies that it is possible to employ fast proximal gradient methods to solve equation 8. We begin with the following lemma that shows that the proximal operator can be determined in terms of the optimal solution of a convex problem that resembles the optimization problem defined in Lemma equation 10 for computing $\mathcal{GS}_k$.

**Lemma 4.1.** *Let $\lambda > 0$, $t \in \mathbb{R}^n$, and $s_1, s_2, \ldots, s_m$ be a set of $m \leq n$ disjoint subsets that partition $[n]$. Then, $v = \text{prox}_{\lambda \mathcal{GS}_k}(t)$ is given by*

$$j \in [m], \quad A_{s_j} v = \frac{u_j A_{s_j} t}{\lambda d_j + u_j}, \tag{13}$$

*where $(u_1, u_2, \ldots, u_n)^T$ is the minimizer of*

$$\min_{\mathbf{u} \in D_k} \sum_{j=1}^{m} \phi\left(\sqrt{d_j} A_{s_j} t, \lambda d_j + u_j\right). \tag{14}$$

Next, we show that the proximal operator of $\mathcal{GS}_k$ reduces to an efficient one-dimensional search.

**Corollary 4.1.1** (The proximal operator of $\mathcal{GS}_k$). *The solution $u_j = u_j(\mu^*)$ of equation 14 with $u_j(\cdot)$ defined as[1]*

$$u(\mu^*) = \begin{cases} 1, & \sqrt{\mu^*} \leq \frac{|b_j|}{\alpha_j + 1}, \\ \frac{|b_j|}{\sqrt{\mu^*}} - \alpha_j, & \frac{|b_j|}{\alpha_j + 1} < \sqrt{\mu^*} < \frac{|b_j|}{\alpha_j}, \\ 0, & \sqrt{\mu^*} \geq \frac{|b_j|}{\alpha_j}. \end{cases} \tag{15}$$

*for $b_j = \left\|\sqrt{d_j} A_{s_j} t\right\|_2$ and $\alpha_j = \lambda d_j (> 0)$, and $\tilde{\eta} = \frac{1}{\sqrt{\mu^*}}$ is a root of the function*

$$g_t(\eta) \equiv \sum_{i=j}^{m} u_j(\eta) - k, \tag{16}$$

*which is nondecreasing and satisfies*

$$g_t\left(\frac{\lambda \cdot \min_{j \in [m]}\{d_j\}}{\left\|\sqrt{d_1} \|A_{s_1} t\|_2, \sqrt{d_2} \|A_{s_2} t\|_2, \ldots, \sqrt{d_m} \|A_{s_m} t\|_2\right\|_\infty}\right) = \sum_{i=1}^{m} 0 - k < 0, \tag{17}$$

*and,*

$$g_t\left(\frac{\lambda \|d_1, d_2, \ldots, d_m\|_\infty + 1}{\left\|M_{\langle s_m \rangle}\left(\sqrt{d_j} t\right)\right\|_2}\right) = \sum_{i=1}^{m} 1 - k > 0. \tag{18}$$

*In addition, $g_t$ can be reformulated as the sum of pairs of the functions*

$$v_i(\eta) \equiv |\eta|b_j| - \alpha_j|, \, w_i(\eta) \equiv 1 - |\eta|b_j| - (\alpha_j + 1)|, j \in [m], \tag{19}$$

*such that,*

$$g_{\mathbf{t}}(\eta) = \frac{1}{2} \sum_{j=1}^{m} v_j(\eta) + \frac{1}{2} \sum_{j=1}^{m} w_j(\eta) - k. \tag{20}$$

The following important remarks are in order.

**Remark 3** (Root search application for function 16). *Employing the randomized root search method in (Beck & Refael, 2022, Algorithm 1) with the $2m$ one break point piece-wise linear functions $v_j, w_j$, as an input to the algorithm, the root of $g_{\mathbf{t}}$ can be found in $O(m)$ time.*

**Remark 4** (Computational complexity of $\text{prox}_{\lambda \mathcal{GS}_k}$). *The computation of $\text{prox}_{\lambda \mathcal{GS}_k}$ boils down to a root search problem (see, Remark 3), which requires $O(m)$ operations. In addition, before employing the root search, the assembly of $v_j, w_j$, requires the calculations of the $m$ values of $b_j$ defined in corollary 16. Note that for any $j \in [m]$ calculating $b_j$ is equivalent to $t^\top A_{s_j} t = \sum_{i \in s_j} t_i^2$. Since $s_j$'s are given, the computational complexity of calculating all $m$ of $b_j$ is linear in $n$, which is the dimension of $t$. Thus, the total computational operations of calculating $\text{prox}_{\lambda \mathcal{GS}_k}$ summarizes to $n$ with is the dimension of all groups parameters together.*

## 5 OPTIMIZATION PROCEDURE

The general problem we are solving is of the form

$$\min_{\mathbf{x} \in \mathbb{R}^n} f(\mathbf{x}) + h(\mathbf{x}), \tag{21}$$

---

[1] If $A_{s_j} t = 0$, then equation 15 implies that $u_i(\mu) = 0$ for all $\mu \geq 0$.

---

**Algorithm 1: General Stochastic Proximal Gradient Method**

---

**Input:** Stepsize $\alpha_t$, $\{\rho_t\}_{t=1}^{t=T} \in [0, 1)$, regularization parameter $\lambda$.
**Initialization:** $\boldsymbol{\theta}_1 \in \mathbb{R}^n$ and $\mathbf{m}_0 = \mathbf{0} \in \mathbb{R}^n$.
**for** iteration $t = 1, \ldots, T$:
    Draw a minibatch sample $\xi_t$
    $\mathbf{g}_t \longleftarrow \nabla f(\boldsymbol{\theta}_t; \xi_t)$
    $\mathbf{m}_t \longleftarrow \rho_t \mathbf{m}_{t-1} + (1 - \rho_t)\mathbf{g}_t$
    $\boldsymbol{\theta}_{t+1} \longleftarrow \mathrm{prox}_{\alpha_t \lambda h}(\boldsymbol{\theta}_t - \alpha_t \mathbf{m}_t)$
**return** $\boldsymbol{\theta}$

---

**Algorithm 2: Learning structured $k$-level sparse neural-network by Prox SGD with WGSEF regularization**

---

**Input:** Stepsize $\alpha_t$, $\{\rho_t\}_{t=1}^{t=T} \in [0, 1)$, regularization parameters $\{\lambda_l\}_{l=1}^{l=L} \in [0, \infty)$.
**Initialization:** Randomly initialize the weights $\theta_{t=0} \in \mathbb{R}^n$, $\mathbf{m}_0 = \mathbf{0} \in \mathbb{R}^n$.
**for** iteration $t = 1, \ldots, T$:
    Draw a minibatch sample $\xi_t$
    $\mathbf{g}_t \longleftarrow \nabla \mathcal{L}(\boldsymbol{\theta}_t; \xi_t)$
    $\mathbf{m}_t \longleftarrow \rho_t \mathbf{m}_{t-1} + (1 - \rho_t)\mathbf{g}_t$
    $\boldsymbol{\theta}_{t+1} \longleftarrow \mathrm{prox}_{\alpha_t \lambda \mathcal{GS}_k}(\boldsymbol{\theta}_t - \alpha_t \mathbf{m}_t)$
**Prune (Optional):** all $\#(m - k)$ smallest $\ell_2$ group norm values of $\boldsymbol{\theta}$.
**return** $\boldsymbol{\theta}$

---

where $f = \frac{1}{N} \sum_{i=1}^{N} f_i : X \to \mathbb{R}$ is continuously differentiable, but possibly nonconvex, and $h$ is a convex function, but possibly nonsmooth. For practical reasons, we cannot store the full gradient $\nabla f(\mathbf{x})$. Hence, we would like to use a stochastic gradient type algorithm. However, such a structure posses several difficulties from an optimization perspective, as most research of stochastic first-order algorithms does not account for both nonconvex smooth term and a nonsmooth convex regularizer. In Ghadimi et al. (2016) the authors provide an analysis of a simple stochastic proximal gradient algorithm, where at each iteration a minibatch of weights is updated using a gradient step followed by a proximal step. This algorithm is proved to converge, however, the rate of convergence depends heavily on the minibatch size, and, in fact, for reasonably sized minibatches it will not converge. J Reddi et al. (2016) proposes variance-reduction type algorithms, but since these extend SAGA Defazio et al. (2014) and SVRG Johnson & Zhang (2013) to the nonconvex and nonsmooth setting, they require storing the gradient for each sample (SAGA) which requires $\mathcal{O}(Nn)$ storage, or recomputing the full gradient every $s \geq N$ iterations (SVRG), which is undesirable for training neural networks. ProxSVGR+ Li & Li (2018) tackles this issue by calculating the gradient for a batch of $|B|$ samples. ProxSVGR+ also has a proven convergence rate, however, this rate is strongly dominated by the magnitude of $|B|$, hence, impractical for training neural networks.

The ProxSGD algorithm Yang et al. (2020) appears appealing to our problem as it allows for momentum. While the algorithm has a convergence guarantee, the authors do not provide the rate, making it less appealing, given the known issue with the minibatch size. We have found the most suitable optimization algorithm to be ProxGen Yun et al. (2021), as it can accommodate momentum, and also has a proven convergence rate with a fixed and reasonable minibatch size of order $\Theta(\sqrt{N})$. Next, we provide a convergence guarantee for Algorithm 1, as given in Yang et al. (2020). This result holds under several regularity assumptions which can be found in Appendix A.3.1.

**Corollary 5.0.1.** *Under Assumptions (C-1)–(C-3) in Appendix A.3.1, Algorithm 1 with constant minibatch size $b_t = b = \Theta(T)$ is guaranteed to yield $\mathbb{E}\left[\mathrm{dist}\left(\mathbf{0}, \widehat{\partial}F(\theta)\right)^2\right] \leq O\left(T^{-1}\right)$, where $\widehat{\partial}F$ is the Fréchet sub-differential function of $F$ (see, Definition 2 in Appendix A.3.1).*

Next, we propose Algorithm 2, as an implementation of Algorithm 1 to solve equation 8, where $f \triangleq \mathcal{L}$ and $h \triangleq \mathcal{GS}_k$. Calculating $\nabla \mathcal{L}(\boldsymbol{\theta}_t; \xi_t)$, commonly approached using a propagation algorithm, is at least linear in the number of parameters and is obviously getting more complex as the number of layers increases. Therefore, the calculation of the prox of the GSFE is not a bottleneck of the update step complexity (since it is linear in the number of parameters). Notice that the reg-

ularization in our setting is layer-separable as can be seen from the definition of the regularization function $h(\mathbf{x}) = \sum_{l=1}^{L} h_l(\mathbf{x}_l)$. Therefore prox is applied to each layer separately (Beck, 2017, Theorem 6.6). Moreover, we allow different $\lambda_l, k_l$ parameters per layer. Accordingly, regularization can be applied differently, namely, not in a per-layer fashion. For example, all convolutional layers can be regularized at once, and groups will be defined so that they create cross-layer sparsity. The only condition is that groups are not overlapping. We note that since Assumptions (C-1)–(C-3) are met, Algorithm 2 converges to an $\epsilon$-stationary point.

Another technique for solving equation 21 is using the HSPG family of algorithms in Chen et al. (2021; 2020); Dai et al. (2023). These algorithms utilize a two-step procedure in which optimization is carried by standard first-order methods (i.e., subgradient or proximal) to find an approximation that is "sufficiently close" to a solution. This step is followed by a half-space step that freezes the sparse groups and applies a tentative gradient step on the dense groups. Over these dense groups, parameters are zeroed-out if a sufficient decrease condition is met, otherwise, a standard gradient step is executed. Notice that the half-space step, as a variant of a gradient method, requires the regularizer term $h$ to have a Lipschitz continuous gradient, which is not satisfied in our setting. However, this property is required only for the dense groups as there is no use of the gradient in groups that are already sparse. Since the continuity is violated only for sparse groups, the condition is satisfied in the required region. Finally, while the "sufficiently close" condition mentioned above cannot be verified in practice, simple heuristics to switch between steps still work well. We can either run the first-order step for a fixed number of iterations before switching to the half-space step, or, alternatively, run the first-order step until the sparsity level stabilizes, and then switch to the half-space step. The dense groups are defined as $\mathcal{I}^0(\mathbf{x}) := \{\gamma \,|\, \gamma \in \mathcal{G}, \|\mathbf{x}^\gamma\| = 0\}$, and sparse groups are defined as $\mathcal{I}^{\neq 0} := \{\gamma \,|\, \gamma \in \mathcal{G}, \|\mathbf{x}^\gamma\| \neq 0\}$. The HSPG pseudo-code (proximal-gradient variant) is given in Algorithm 3 in the appendix. We mention the enhanced variant of the standard HSPG algorithm named AdaHSPG+ Dai et al. (2023) improves upon that implementing adaptive strategies that optimize performance, focusing on better handling of complex or dynamic problem scenarios where standard HSPG may be less efficient.

## 6    EXPERIMENTS

We provide only a summary of our experimental results here, deferring full details to Appendix A.5. To demonstrate the performance of prox-SGD with WSEG (Algorithm 2), in terms of both compression and accuracy, we utilized widely-recognized benchmark DNN architectures: VGG16 Simonyan & Zisserman (2014), ResNet18 He et al. (2016a), and MobileNetV1 Howard et al. (2017). These architectures were tested on the datasets CIFAR10 Krizhevsky et al. (2009) and Fashion-MNIST Xiao et al. (2017). All experiments were conducted over 300 epochs. For the first 150 epochs, we employed Algorithm 2, and for the subsequent epochs, we used the HSPG with the WGSEF as a regularize (Algorithm 3). Experiments were conducted using a mini-batch size of $b = 128$ on an A100 GPU. The coefficient for the WGSE regularizer was set to $\lambda = 10^{-2}$. In Table 1, we compare our results with those reported in Dai et al. (2023). The primary metrics of interest are the neural network group sparsity ratio and prediction accuracy (Top-1). Notably, the WGSE achieves a markedly higher group sparsity compared to all other methods except AdaHSPG+, for which we obtained slightly higher results. It should be mentioned that all techniques achieved comparable generalization error rates on the validation datasets. We mention here that all the optimization methods in Table 1 used the well-known Group Lasso as a regularizer, because as observed in Bui et al. (2021), it gave the best results, in terms of compression and performance, in comparison to many other convex and nonconvex regularizers.

Next, we examine the effectiveness of the WGSEF in the LeNet-5 convolutional neural network LeCun et al. (1998), on the MNIST dataset LeCun & Cortes (2010). The networks were trained without any data augmentation. We apply the WGSEF regularization on filters in convolutional layers using a predefined value for the sparsity level $k$. Table 2 summarizes the number of remaining filters at convergence, floating-point operations (FLOP), and the speedups. We evaluate these metrics both for a LeNet-5 baseline (i.e., without sparsity learning), and our WGSEF sparsification technique. To be accurate and fair in comparison, the baseline model was trained using SGD. It can be seen that WGSEF reduces the number of filters in the convolution layers by a factor of the half, as dictated by $k = 8$, while the accuracy level did not decrease. Furthermore, since the sparsification is structural, there is a significant improvement in flops, as well as in the latency time of inference.

Table 1: Comparison of state-of-the-art techniques to our WGSEF technique, in terms of group sparsity ratio/validation accuracy in percentage, for various models and datasets.

| Model | Dataset | Prox-SG | Prox-SVRG | HSPG | AdaHSPG+ | WGSEF |
|-------|---------|---------|-----------|------|----------|-------|
| VGG16 | CIFAR10 | 54.0/90.6 | 14.7/89.4 | 74.6/91.1 | 76.1/91.0 | **76.8/91.5** |
| | F-MNIST | 19.1/93.0 | 0.5/92.7 | 39.7/**93.0** | 51.2/92.9 | **51.9**/92.8 |
| ResNet18 | CIFAR10 | 26.5/94.1 | 2.8/94.2 | 41.6/94.4 | 42.1/**94.5** | **42.6/94.5** |
| | F-MNIST | 0.0/94.8 | 0.0/94.6 | 10.4/**94.9** | 43.9/**94.9** | **44.2/94.9** |
| MobileNetV1 | CIFAR10 | 58.1/91.7 | 29.2/90.7 | 65.4/**92.0** | 71.5/91.8 | **71.8**/ 91.9, |
| | F-MNIST | 62.6/94.2 | 42.0/94.2 | 74.3/94.5 | 78.9/**94.6** | **79.1**/94.5 |

Repeating the same experiment, but now constraining the number of non-pruned filters in the second convolutional layer to be at most 4 (i.e., at most quarter of the baseline), the accuracy slightly deteriorates; however, significant improvements can be observed in both the flops number and the speed up, as expected. The networks were trained with a learning rate of $0.001$, regularization magnitude $\lambda = 10^{-5}$, and a batch size of 32 for 150 epochs across 5 runs.

Table 2: Results of running algorithm 2, onto redundant filters in LeNet (in the order of conv1-conv2).

| LeNet (MNIST) | Error | Filter (sparsity-level) | FLOP | Speedup |
|---------------|-------|------------------------|------|---------|
| Baseline (SGD) | 0.84 % | 6-16 | 100 %-100 % | 1.00 ×-1.00 × |
| WGSEF | 0.78 % | 3-8 | 48.7 %-21.6 % | 2.06×-4.53 × |
| WGSEF | 0.89 % | 3-4 | 48.7 %-14.7 % | 2.06×-7.31 × |
| LeNet (MNIST) | Error | Parameters | FLOP | Speedup |
| Unstructured WGSEF | 0.76 % | 75(/150)-1200(/2400) | 68.7 %-59.2 % | 1×-1× |

We also implemented deep residual networks (ResNet50) He et al. (2016b) and trained it on CIFAR-10 Krizhevsky et al. (2009), applying WGSEF regularization with a predefined sparsity level that is at most equal to half the number of filters at each of the convolutional layers. Again, to be accurate in comparison, the baseline model was trained using SGD, both with an initial learning rate of $\alpha_0 = 0.01$, regularization magnitude $\lambda = 0.3$, a batch size of 128, and Cosine Annealing learning rate scheduler. The results are similar in nature to those of the MNIST experiment in Table 3.

Table 3: Average results running algorithm 2 on ResNet50 for CIFAR-10 over 5 trials, after completing 110 epochs, using parameters $\alpha_0 = 0.01$ and $\lambda = 0.3$.

| Resnet50 (CIFAR-10) | Error | FLOP | Speedup |
|---------------------|-------|------|---------|
| Baseline (SGD) | 7.62 % | 100 % | 1.00 × |
| WGSEF | 7.4 % | 48.7 % | 2.06 × |

Finally, table 6 presents the results of training both VGG16 and DenseNet40 Huang et al. (2017) on CIFAR100 Krizhevsky et al. (2009), while applying WGSEF regularization with a predefined sparsity level of half the number of channels for VGG16, and $60\%$ of those of the DenseNet40. The baseline model was trained using SGD, both with an initial learning rate of $\alpha = 0.01$ and regularization magnitude $\lambda = 0.01$.

Table 4: Results of running algorithm 2, onto redundant Channels on CIFAR100, over 250 epochs.

| DCNN | Model | Error (%) | Pruned Channels | Overall Density |
|------|-------|-----------|-----------------|-----------------|
| VGG16 | Baseline-unpruned | 26.28 | - | - |
| | WGSEF | 26.46 | 50% | 41.3% |
| DenseNet40 | Baseline-unpruned | 25.36 | - | - |
| | WGSEF | 25.6 | 60% | 42.8% |

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

# A APPENDIX

## A.1 PROOFS FOR SECTION 3

### A.1.1 PROOF FOR LEMMA 3.1

*Proof.* Let us define the axillary diagonal positive definite matrix $D^{n \times n}$, where the $D_{i,i}$ entry holds $D_{i,i} = \sqrt{d_i}, \forall i \in s_j, d_i = d_j$. Now, consider the following chain of equalities:

$$gs_k^\star(\tilde{\theta}) = \max_{\theta \in \mathbb{R}^n} \{\langle \tilde{\theta}, \theta \rangle - gs_k(\theta)\}$$

$$= \max_{\theta \in \mathbb{R}^n} \left\{ \tilde{\theta}^\top \theta - \frac{1}{2} \sum_{j=1}^m d_j \|\theta s_j\|_2^2 - \delta_{C_k}(\theta) \right\}$$

$$= \max_{\substack{\theta \in C_k \\ \forall i \in s_j, d_i = d_j}} \left\{ \tilde{\theta}^\top \theta - \sum_{j=1}^m d_j \cdot \|\theta s_j\|_2^2 \right\}$$

$$= \max_{\substack{\theta \in C_k \\ \forall i \in s_j, d_i = d_j}} \left\{ \tilde{\theta}^\top \theta - \frac{1}{2} \theta^\top D^\top D \theta \right\}$$

$$\overset{(a)}{=} \max_{\substack{t \in \tilde{C}_k \\ \forall i \in s_j, d_i = d_j}} \left\{ \tilde{\theta}^\top D^{-1} t - \frac{1}{2} t^\top t \right\}$$

$$= \max_{\substack{t \in \tilde{C}_k \\ \forall i \in s_j, d_i = d_j}} \left\{ -\frac{1}{2} \|t - D^{-1}\tilde{\theta}\|_2^2 + \frac{1}{2} \|D^{-1}\tilde{\theta}\|_2^2 \right\}$$

$$= \frac{1}{2} \|D^{-1}\tilde{\theta}\|_2^2 + \max_{\substack{t \in \tilde{C}_k \\ \forall i \in s_j, d_i = d_j}} \left\{ -\frac{1}{2} \|t - D^{-1}\tilde{\theta}\|_2^2 \right\}$$

$$= \frac{1}{2} \|D^{-1}\tilde{\theta}\|_2^2 + \max_{\substack{t \in \tilde{C}_k \\ \forall i \in s_j, d_i = d_j}} \left\{ -\frac{1}{2} \sum_{i=1}^n (t_i - (D^{-1})_{ii} \tilde{\theta}_i)^2 \right\}$$

$$\overset{(b)}{=} \frac{1}{2} \|D^{-1}\tilde{\theta}\|_2^2 + \max_{\substack{t \in \tilde{C}_k \\ \forall i \in s_j, d_i = d_j}} \left\{ -\frac{1}{2} \sum_{j=1}^m \left\| M_{s_j}(t - D^{-1}\tilde{\theta}) \right\|_2^2 \right\}$$

$$= \frac{1}{2} \|D^{-1}\tilde{\theta}\|_2^2 - \frac{1}{2} \sum_{j=m-k}^m \left\| M_{\langle s_j \rangle}(D^{-1}\tilde{\theta}) \right\|_2^2$$

$$\overset{(c)}{=} \frac{1}{2} \sum_{j=1}^k \left\| M_{\langle s_j \rangle}(D^{-1}\tilde{\theta}) \right\|_2^2$$

$$= \frac{1}{2} \sum_{j=1}^k \frac{1}{d_j} \left\| M_{\langle s_j \rangle}(\tilde{\theta}) \right\|_2^2$$

where, $(a)$ the set $\tilde{C}_k$ is given by

$$\tilde{C}_k = \left\{ \theta : \left\| \|(D\theta) s_1\|^2, \|(D\theta) s_2\|^2 \dots, \|(D\theta) s_m\|^2 \right\|_0 \le k \right\},$$

$(b)$ follows by the fact that the sum of the squares of the coordinates of the input vector in the support of the disjoint subset $s_1, s_2, \ldots, s_m$ that completes the index space $\bigcup_{i=1}^m s_i = [n]$ is equal to the sum of squares of the coordinates of the original vector, and $(c)$ follows by the fact that $\sum_{j=m-k}^m \left\| M_{\langle s_j \rangle}(D^{-1}\tilde{\theta}) \right\|_2^2$ is the sum of square $\ell_2$-norm of the $m - k$ disjoint subsets of $D^{-1}\tilde{\theta}$ with the smallest $\ell_2$-norm, while is the sum of squared $\ell_2$-norm of all disjoint subsets of $D^{-1}\tilde{\theta}$. $\quad\square$

### A.1.2 PROOF FOR LEMMA 3.2

*Proof.* We first note that

$$\sum_{i=1}^{k} \left\| M_{\langle s_i \rangle}(D^{-1}\tilde{\theta}) \right\|_2 = \max_{u \in B_k} \sum_{i=1}^{m} u_i \left\| M_{s_i}(D^{-1}\tilde{\theta}) \right\|_2^2, \tag{22}$$

where

$$B_k \triangleq \left\{ u \in \mathbb{R}^m \mid 0 \leqslant u \leqslant e, e^\top u \leq k \right\}. \tag{23}$$

Now, Consider the following chain of inequalities:

$$
\begin{aligned}
\mathcal{GS}_k(\theta) &= \max_{\tilde{\theta} \in \mathbb{R}^n} \left\{ \langle \theta, \tilde{\theta} \rangle - g s^\star(\tilde{\theta}) \right\} \\
&= \max_{\tilde{\theta} \in \mathbb{R}^n} \left\{ \theta^\top \tilde{\theta} - \frac{1}{2} \sum_{i=1}^{k} \left\| M_{\langle s_i \rangle}(D^{-1}\tilde{\theta}) \right\|_2^2 \right\} \\
&= \max_{\substack{\tilde{\theta} \in \mathbb{R}^n \\ t = D^{-1}\tilde{\theta}}} \left\{ \theta^\top D t - \frac{1}{2} \sum_{i=1}^{k} \left\| M_{\langle s_i \rangle}(t) \right\|_2^2 \right\} \\
&= \max_{t \in \mathbb{R}^n} \left\{ \theta^\top D t - \frac{1}{2} \max_{u \in D_k} \sum_{i=1}^{m} u_i \left\| M_{s_i}(t) \right\|_2^2 \right\} \\
&= \max_{t \in \mathbb{R}^n} \left\{ \theta^\top D t - \frac{1}{2} \max_{u \in D_k} \left\{ \sum_{j=1}^{m} u_j \left( t^\top A_{s_j}^\top A_{s_j} t \right) \right\} \right\} \\
&\overset{(a)}{=} \max_{t \in \mathbb{R}^n} \left\{ \theta^\top D t + \frac{1}{2} \min_{\mathbf{u} \in D_k} \left\{ \sum_{j=1}^{m} (-u_j) \left( t^\top A_{s_j} t \right) \right\} \right\} \\
&= \frac{1}{2} \max_{t \in \mathbb{R}^n} \left\{ \min_{\mathbf{u} \in D_k} \left\{ 2\theta^\top D t - \sum_{j=1}^{m} u_j t^\top A_{s_j} t \right\} \right\} \\
&\overset{(b)}{=} \frac{1}{2} \min_{\mathbf{u} \in D_k} \left\{ \max_{t \in \mathbb{R}^n} \left\{ 2\theta^\top D t - \sum_{j=1}^{m} u_j t^\top A_{s_j} t \right\} \right\} \\
&= \frac{1}{2} \min_{\mathbf{u} \in D_k} \left\{ \sum_{j=1}^{m} \max_{t \in \mathbb{R}^n} 2\theta^\top A_{s_j} D t - u_j t A_{S_j} t \right\} \\
&= \frac{1}{2} \min_{\mathbf{u} \in D_k} \left\{ \sum_{j=1}^{m} \max_{t \in \mathbb{R}^n} 2\sqrt{d_j} \theta^\top A_{s_j} t - u_j t A_{S_j} t \right\} \\
&= \frac{1}{2} \min_{\mathbf{u} \in D_k} \left\{ \sum_{j=1}^{m} \phi \left( \sqrt{d_j} A_{s_j} \theta, u_j \right) \right\} \tag{24} \\
&= \frac{1}{2} \min_{\mathbf{u} \in D_k} \left\{ \sum_{j=1}^{m} d_j \phi \left( A_{s_j} \theta, u_j \right) \right\},
\end{aligned}
$$

where $(a)$ follows by the fact that $A_{s_j}$ is a self-adjoint matrix, and $(b)$ follows from the fact that the objective function is concave w.r.t. $\tilde{y}$ and convex w.r.t $u$, and the MinMax Theorem v. Neumann (1928). $\qquad\square$

### A.1.3 PROOF FOR COROLLARY 3.2.1

*Proof.* Directly by expression (34). $\qquad\square$

## A.2 Proofs for section 4

### A.2.1 Proof of lemma 4.1

*Proof.* Recall that

$$v = \text{prox}_{\lambda \mathcal{GS}_k}(t) = \underset{\theta \in \mathbb{R}^n}{\text{argmin}} \left\{ \lambda \mathcal{GS}_k(\theta) + \frac{1}{2} \|\theta - t\|_2^2 \right\}.$$

Using Lemma 3.2, the above minimization problem can be written as

$$\min_{\mathbf{u} \in D_k} \min_{\theta \in \mathbb{R}^n} \left\{ \Phi(\theta, u, t) \equiv \frac{\lambda}{2} \sum_{j=1}^m d_j \phi \left( A_{s_j} \theta, u_j \right) + \frac{1}{2} \|\theta - t\|_2^2 \right\}$$

$$= \min_{\mathbf{u} \in D_k} \min_{\theta \in \mathbb{R}^n} \left\{ \Phi(\theta, u, t) \equiv \frac{\lambda}{2} \sum_{j=1}^m d_j \phi \left( A_{s_j} \theta, u_j \right) + \frac{1}{2} M_{s_j} (\theta - t)_2^2 \right\}$$

$$= \min_{\mathbf{u} \in D_k} \min_{\theta \in \mathbb{R}^n} \left\{ \Phi(\theta, u, t) \equiv \frac{\lambda}{2} \sum_{j=1}^m d_j \phi \left( A_{s_j} \theta, u_j \right) + \frac{1}{2} (\theta - t)^\top A_{s_j} (\theta - t) \right\}. \quad (25)$$

Solving for $\theta$, we get that for any $j \in [m]$, if $d_j A_{s_j} \theta \geq 0$ then,

$$\frac{d_j \lambda A_{s_j} \hat{\theta}}{u_j} + A_{s_j} (\hat{\theta} - t) = 0$$

$$A_{s_j} \hat{\theta} \left( \frac{d_j \lambda}{u_j} + 1 \right) - A_{s_j} t = 0$$

$$A_{S_j} \left( \hat{\theta} \left( \frac{d_j \lambda}{u_j} + 1 \right) - t \right) = 0,$$

meaning that,

$$v_i = \frac{t_i u_j}{\lambda d_j + u_j}, \; j \in [m], i \in s_j, \quad (26)$$

or, equivalently,

$$A_{s_j} v = \frac{u_j A_{s_j} t}{\lambda d_j + u_j}, \; j \in [m]. \quad (27)$$

Next, we show that $u$ is the minimizer of the problem $\min_{\mathbf{u} \in D_k} \Phi(\theta, u, t)$. Equation equation 27 also holds when $u_j = 0$, since in that case, $v_i = \hat{\theta}_i = 0$, for all $i \in s_j$. Plugging equation 27 in $\Phi$, yields,

$$\Phi(\hat{\theta}, u, t) = \frac{1}{2} \sum_{j=1}^m d_j \left( \lambda \frac{\hat{\theta}^\top A_{s_j} \hat{\theta}}{u_i} \right) + \frac{1}{2} \|\hat{\theta} - \mathbf{t}\|_2^2 \quad (28)$$

$$= \frac{1}{2} \sum_{j=1}^m \left( \lambda d_j \frac{\hat{\theta}^\top A_{s_j} \hat{\theta}}{u_j} + \|A_{s_j} \left( \hat{\theta} - t \right) \|_2^2 \right)$$

$$= \frac{1}{2} \sum_{j=1}^m \left( \lambda d_j \frac{u_j^2 t^\top A_{s_j} t}{u_j (\lambda d_j + u_j)^2} + \left\| \frac{\lambda d_j A_{s_j} t}{\lambda d_j + u_j} \right\|_2^2 \right)$$

$$= \frac{1}{2} \sum_{j=1}^m \left( \lambda d_j \frac{u_j t^\top A_{s_j} t}{(\lambda d_j + u_j)^2} + \frac{(\lambda d_j)^2 t^\top A_{s_j} t}{(\lambda d_j + u_j)^2} \right)$$

$$= \frac{\lambda}{2} \sum_{j=1}^m d_j \frac{t^\top A_{s_j} t}{\lambda d_j + u_j}$$

$$= \frac{\lambda}{2} \sum_{j=1}^m \phi \left( \sqrt{d_j} A_{s_j} t, \lambda d_j + u_j \right), \quad (29)$$

which concludes the proof. □

A.2.2 PROOF OF COROLLARY 4.1.1

*Proof.* Assigning a Lagrange multiplier for the inequality constraint $\mathbf{e}^T\mathbf{u} \leq k$ in problem (25), we obtain the Lagrangian function

$$L(\mathbf{u}, \mu) = \sum_{j=1}^{m} \left( \phi\left(\sqrt{d_j}A_{s_j}t, \lambda d_j + u_j\right) + \mu u_j \right) - k\mu.$$

Therefore, the dual objective function is given by

$$q(\mu) \equiv \min_{\mathbf{u}:0\leq\mathbf{u}\leq\mathbf{e}} L(\mathbf{u}, \mu) = \sum_{j=1}^{m} \varphi_{b_j,\alpha_j}(\mu) - k\mu, \text{ for, } b_j = \left\|\sqrt{d_j}A_{s_j}t\right\|_2 \text{ and } \alpha_j = \lambda d_j (> 0).$$

(30)

where for any $b \in \mathbb{R}$ and $\alpha \geq 0$, the function $\varphi_{b,\alpha}$ is defined in Beck & Refael (2022) by

$$\varphi_{b,\alpha}(\mu) \equiv \min_{0\leq u\leq 1}\{\phi(b, \alpha + u) + \mu u\}, \quad \mu \geq 0.$$

(31)

Thus, the dual of problem (14) is the maximization problem

$$\max\{q(\mu) : \mu \geq 0\}$$

(32)

A direct projection of Lemma (Beck & Refael, 2022, lemma 2.4) is that if $\tilde{\mu} > 0$, the function $\mathbf{u} \mapsto L(\mathbf{u}, \tilde{\mu})$ has a unique minimizer over $\{\mathbf{u} \in \mathbb{R}^m : \mathbf{0} \leq \mathbf{u} \leq \mathbf{e}\}$ given by $u_j = \varphi'_{b_j,\alpha_j}(\tilde{\mu})$, where it was shown that

$$\varphi_{b_j,\alpha_j}(\mu) = \begin{cases} \frac{b_j^2}{\alpha_j+1} + \mu, & \sqrt{\mu} \leq \frac{|b_j|}{\alpha_j+1} \\ 2|b_j|\sqrt{\mu} - \alpha_j\mu, & \frac{|b_j|}{\alpha_j+1} < \sqrt{\mu} < \frac{|b_j|}{\alpha_j}, \\ \frac{b^2}{\alpha_j}, & \sqrt{\mu} \geq \frac{|b_j|}{\alpha_j}, \end{cases}$$

for $b > 0$, otherwise 0, and the minimizer is given by

$$u(\mu^*) = \begin{cases} 1, & \sqrt{\mu} \leq \frac{|b_j|}{\alpha_j+1}, \\ \frac{|b_j|}{\sqrt{\mu}} - \alpha_j, & \frac{|b_j|}{\alpha_j+1} < \sqrt{\mu} < \frac{|b_j|}{\alpha_j}, \\ 0, & \sqrt{\mu} \geq \frac{|b_j|}{\alpha_j}. \end{cases}$$

Problem (32), is concave differentiable and thus the minimizer $\tilde{\mu}$ holds $q'(\tilde{\mu}) = 0$, meaning

$$q'(\tilde{\mu}) = \sum_{j=1}^{m} u_j(\mu) - k = 0.$$

We observe that for any $j \in [m]$ the functions $u_j(\mu)$ are monotonically continuous nonincresing, and therefore utilizing Lemma (Beck & Refael, 2022, lemma 3.1) $\mu^* = \frac{1}{\eta^2}$ is the a root of the nondecreasing function,

$$g_t(\eta) \equiv \sum_{j=1}^{m} u_j(\eta) - k.$$

Note that for

$$g_t\left(\frac{\lambda \cdot min_{j\in[m]}\{d_j\}}{\left\|\sqrt{d_1}\|A_{s_1}t\|_2, \sqrt{d_2}\|A_{s_2}t\|_2, \ldots, \sqrt{d_m}\|A_{s_m}t\|_2\right\|_\infty}\right) = \sum_{i=1}^{m} 0 - k < 0,$$

while

$$g_t\left(\frac{\lambda\|d_1, d_2, \ldots, d_m\|_\infty + 1}{\left\|M_{\langle s_m\rangle}\left(\sqrt{d_j}t\right)\right\|_2}\right) = \sum_{i=1}^{m} 1 - k > 0.$$

Now, applying lemma (Beck & Refael, 2022, lemma 3.2), we deduce that $u_j(\mu)$, can be divided into the sum of the two following functions,

$$v_j(\eta) \equiv |\eta|b_j| - \alpha_j|, w_j(\eta) \equiv 1 - |\eta|b_j| - (\alpha_j + 1)|, j \in [m],$$

and thus $g_t$ can be reformulated as follows

$$g_\mathbf{t}(\eta) = \frac{1}{2}\sum_{j=1}^{m} v_j(\eta) + \frac{1}{2}\sum_{j=1}^{m} w_j(\eta) - k.$$

$\square$

### A.3 PROOFS FOR SECTION 5

#### A.3.1 ASSUMPTIONS

Consider the following definition needed to present our convergence result.

**Definition 2** (Fréchet Subdifferential). *Let $\varphi$ be a real-valued function. The Fréchet subdifferential of $\varphi$ at $\bar{\theta}$ with $|\varphi(\bar{\theta})| < \infty$ is defined by*

$$\widehat{\partial}\varphi(\bar{x}) \triangleq \left\{ \theta^* \in \Omega \ \middle| \ \liminf_{\theta \to \bar{\theta}} \frac{\varphi(\theta) - \varphi(\bar{\theta}) - \langle \theta^*, \theta - \bar{\theta} \rangle}{\|\theta - \bar{\theta}\|} \geq 0 \right\}. \tag{33}$$

The following are the assumptions under which Corollary 5.0.1 holds.

**(C-1)** ($L$-smoothness) The loss function $f$ is differentiable, $L$-smooth, and lower-bounded,
$$\|\nabla f(x) - \nabla f(y)\| \leq L\|x - y\| \quad \text{and} \quad f(x^*) > -\infty.$$

**(C-2)** (Bounded variance) The stochastic gradient $g_t = \nabla f(\theta_t; \xi)$ is unbiased and,
$$\mathbb{E}_\xi\big[\nabla f(\theta_t; \xi)\big] = \nabla f(\theta_t), \quad \mathbb{E}_\xi\big[\|g_t - \nabla f(\theta_t)\|^2\big] \leq \sigma^2.$$

**(C-3)** (i) Final step-vector is finite, (ii) the stochastic gradient is bounded, and (iii) the momentum parameter is exponentially decaying, namely,

(i) $\|\theta_{t+1} - \theta_t\| \leq D$,     (ii) $\|g_t\| \leq G$,     (iii) $\rho_t = \rho_0 \mu^{t-1}$,

with $D, G > 0$ and $\rho_0, \mu \in [0, 1)$.

#### A.3.2 PROOF OF COROLLARY 5.0.1

*Proof.* The proof follows from (Yun et al., 2021, Corollary 1), by taking $C_t = 0$ and $\delta = 1$.    $\square$

### A.4 ADDITIONAL ALGORITHM

---
**Algorithm 3: General Stochastic Proximal Gradient Method**

---
**Input:** Stepsize $\alpha_t$, $\{\rho_t\}_{t=1}^{t=T} \in [0, 1)$, regularization parameter $\lambda$, switch condition $\mathcal{S}$, projection threshold $\epsilon$.
**Initialization:** $\boldsymbol{\theta}_1 \in \mathbb{R}^n$ and $\mathbf{m}_0 = \mathbf{0} \in \mathbb{R}^n$.
**for** iteration $t = 1, \dots, T$:
  **if** condition $\mathcal{S}$ is **not** satisfied:
    Apply Algorithm 2
  **else**:
    Draw a minibatch sample $\xi_t$
    $\mathbf{g}_t \longleftarrow \nabla f(\boldsymbol{\theta}_t^{\mathcal{I}^{\neq 0}}; \xi_t) + \nabla h(\boldsymbol{\theta}_t^{\mathcal{I}^{\neq 0}}; \xi_t)$
    $\tilde{\boldsymbol{\theta}}_t^{\mathcal{I}^{\neq 0}} \longleftarrow \boldsymbol{\theta}_{t-1} - \alpha_t \mathbf{g}_t$, $\tilde{\boldsymbol{\theta}}_t^{\mathcal{I}^0} \longleftarrow \mathbf{0}$
    **for** each group $\gamma \in \mathcal{I}^{\neq 0}$:
      **if** $\langle \tilde{\boldsymbol{\theta}}_t^\gamma, \boldsymbol{\theta}_{t-1}^\gamma \rangle < \epsilon \|\boldsymbol{\theta}_{t-1}^\gamma\|^2$:
        $\tilde{\boldsymbol{\theta}}_t^\gamma \longleftarrow \mathbf{0}$
    $\boldsymbol{\theta}_{t+1} \longleftarrow \tilde{\boldsymbol{\theta}}_t^\gamma$
**return** $\theta$

---

### A.5 ADDITIONAL EXPERIMENTS

We examine the effectiveness of the WGSEF in the LeNet convolutional neural network LeCun et al. (1998) (the architecture is given in the appendices), on the FasionMNIST dataset. The networks were trained without any data augmentation. We apply the WGSEF regularization on filters in convolutional layers using a predefined value for the sparsity level $k$. Table 5 summarizes the number of remaining filters at convergence, floating-point operations (FLOP), and the speedups. We evaluate these metrics both for a LeNet baseline (i.e., without sparsity learning), and our WGSEF sparsification technique. To be accurate and fair in comparison, the baseline model was trained using SGD. We use a learning rate equal to $1e - 4$, with a batch size of 32, a momentum 0.95, and 15 epochs.

Table 5: Results of training while applying WGSEF sparsification (with $\lambda = 0.05$), onto redundant filters in LeNet (in the order of conv1-conv2), and neurons in Linear layers.

| LeNet (FasionMNIST) | Error | Filter (non-sparse) | Linear layers sparsity | Speedup (conv1-conv2) |
|---|---|---|---|---|
| Baseline (SGD) | 11.1 % | 6-16 | 0 % | 1.00 ×-1.00× |
| WGSEF | 11 % | 3-8 | 0% | 2×-4.5× |
| WGSEF | 14 % | 4-6 | 62% | 1.7×-6.1× |
| WGSEF | 12.3 % | 2-3 | 0% | 2×-7.12× |

The second row of Table 5 shows that the method resulted in a significant decrease in the number of non-zero filters, in this case, half of the filters (groups of parameters) were zeroed while still improving the model performance. The rest of the experiments show that there was a higher sparsification in the number of non-zero filters (groups of parameters), with only a negligible degradation in the model's accuracy.

Following, in figure 1, the two graphs refer to the training procedure of the model corresponding with the last row of Table 5. These graphs illustrate the sparsity of the model parameters (in percentages) and the corresponding accuracy as a function of the epoch number. Notably, the desired sparsity level was rapidly attained within the first three epochs. Subsequently, the model enhanced its accuracy throughout the remaining epochs without compromising the achieved sparsity.

Figure 1: The graph on the left shows the level of sparseness (in percentages) as a function of epoch number, while the graph on the right shows the model's accuracy as a function of epoch number.

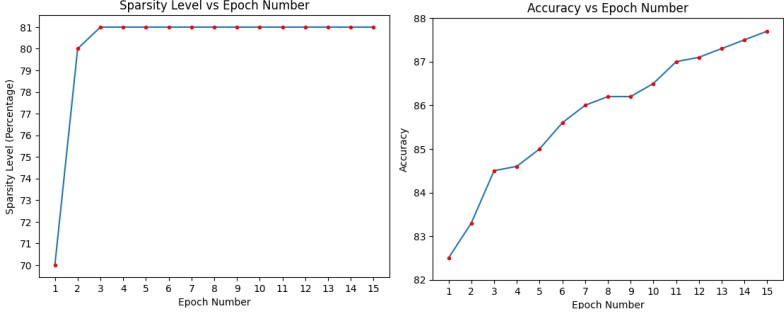

This shows that Algorithm 2 results in the actual zeroing of the (groups of) parameters intended for pruning, and not just pushing them close to zero, hence practically the last line in Algorithm 2 which proposed as optional, is actually barely needed.

Figure 2: Ratio of the sparse filters in the convolutional layers, according to the order of the layers in the resnet18 model, as obtained by Algorithm 2, and corresponds to the experiment in row 3 of table 2.

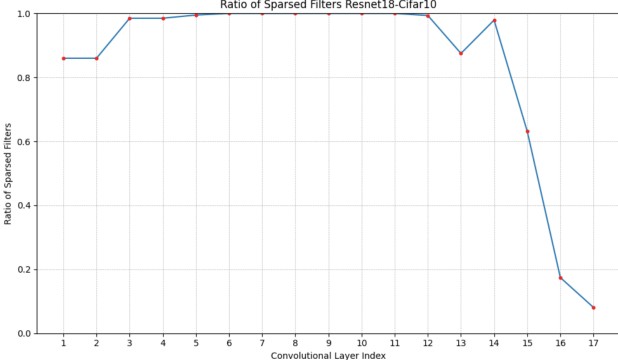

## A.6 LeNet convolutional neural network Architecture

| Layer | # filters I neurons | Filter size | Stride | Size of feature map | Activation function |
|---|---|---|---|---|---|
| Input | – | – | – | $32 \times 32 \times 1$ | |
| Conv 1 | 6 | $5 \times 5$ | 1 | $28 \times 28 \times 6$ | Relu |
| MaxPool2d | | $2 \times 2$ | 2 | $14 \times 14 \times 6$ | |
| Conv 2 | 16 | $5 \times 5$ | 1 | $10 \times 10 \times 16$ | Relu |
| MaxPool2d | | $2 \times 2$ | 2 | $5 \times 5 \times 16$ | |
| Fully Connected 1 | – | – | – | 120 | Relu |
| Fully Connected 2 | – | – | – | 84 | Relu |
| Fully Connected 3 | – | – | – | 10 | Softmax |

