# OpenReview forum: "Learning Structured Sparse Neural Networks Using Group Envelope Regularization"
_ICLR.cc/2024/Conference — ICLR 2024 Conference Withdrawn Submission_

### Official Review · Reviewer_kcJf · 2023-10-31

**Soundness:** 3 good
**Presentation:** 2 fair
**Contribution:** 3 good
**Rating:** 5
**Confidence:** 5

**Summary:**

The paper introduces a regularizer called Weighted Group Sparse Envelope Function that is used with a proximal gradient type of algorithm to sparsify neural networks.

**Strengths:**

Addresses the important problem of sparsifying neural networks.

**Weaknesses:**

The literature review only briefly touches previous works that directly optimize the L0 penalty and only cites old papers from before 2017 and omits more recent works in this direction. See below for one example.

The description of the method has problems, raising reproducibility issues:
- It is unnecessarily complicated and could confuse the reader, For example, in eq (13) theta^T A_s_j theta is just the L2 norm of the theta on index set s_j.
- It has undefined terms. For example SEF is referred in the introduction as the sparse envelope function, but its definition is never given, but is referred to as s_k^**  in Remark 1 and later, and s_k^** is never defined.

The paper is missing a Conclusion section and a discussion of the weaknesses of the proposed method.

Experimental evaluation is lacking in many respects:
- It is not clear what the AdaHSPG+ algorithm is because it is not defined.
- Experiments with comparisons are done only on two easy datasets. More challenging datasets such as CIFAR-100 or Imagenet are not included.
- The comparison is only with group Lasso based methods, relying on a statement from a Bui et al 2021 paper claiming that they work best. No works after 2021 that are not based on group Lasso are included in the evaluation. For example, Guo et al "Network pruning via annealing and direct sparsity control", IJCNN 2021 obtains better results on CIFAR-10 with VGG16.

**Questions:**

- How does the proposed method perform on CIFAR-100?
- Lasso based methods are known to bias the weights while imposing sparsity, obtaining suboptimally trained NNs. How is that avoided in the proposed approach?
- How are the regularization parameters lambda_l chosen to agree with the desired sparsity k?

---

> ### Author Response · Authors · 2023-11-23
> **Reply to reviewer kcJf**
>
> We are pleased to address the points raised by reviewer kcJf.
>
> 1.	We would like to draw the attention of the reviewer that $\theta^T A_{s_j}$ (as in equation) *is not* the L2 norm of the theta on index set $s_j$, but rather a vector where every entry which its index belongs to $s_j$ equals to its corresponding in $\theta$, and otherwise zero.
> 2.	We would like to draw the attention of the reviwer that $s_k$ was defined in Remark 1s_k(\theta) = \frac{1}{2}\|{\theta}\|_{2}^{2}$ if $\|{\theta}\|_{0}\leq k$ and $s_k(\theta)=\infty$, otherwise.
> 3.	Thanks for the author for his comment, we give a description of AdaHSPG+ algorithm and cition in section 5.
> 4.	We have added additional results for cifar100.
> 5.	Lasso works best when the underlying true model is sparse. In many real-world neural network applications, especially in complex datasets, this assumption of sparsity might not hold true. Hence, applying Lasso could lead to an oversimplified model that fails to capture the complexity of the data.
> On the one hand, we know that methods of this type create a built-in bias, but on the other hand, they reduce the model variance, and thus the performance increases. This has been proven in linear models (linear regression) and shown empirically in deep learning models. Since the regularizer we propose embodies prior knowledge about the model suitable for the problem, i.e. it is over-parameterized and therefore has redundant components, then we can assume that the limited search space, the one that will lead to a bias in the model, is also the one that will necessarily reduce the variance of the model.
> 6.	There is interchangeability between K and Lambda, but an advantage of WGSEF is that we only really need to control Lambda. K should be set so it represents the desired sparsity level, and for reasonable values of Lambda it will be a lower bound on the sparsity level. The user only needs to adjust Lambda, which is no different than Lasso / Group Lasso.

---

### Official Review · Reviewer_FGZh · 2023-11-02

**Soundness:** 2 fair
**Presentation:** 1 poor
**Contribution:** 2 fair
**Rating:** 3
**Confidence:** 3

**Summary:**

The paper presents a regularization approach, the Weighted Group Sparse Envelope Function (WGSEF), for inducing structured sparsity in neural networks, facilitating hardware-accelerated deep learning. It introduces an efficient calculation method for WGSEF and a proximal-gradient optimization technique to train sparse models. The paper's experimental results demonstrate the method's ability to maintain or improve accuracy while achieving computational efficiency.

**Strengths:**

- The paper introduces WGSEF, a novel generalization of the sparse envelope function for structured sparsity.

- It provides an efficient technique for the calculation of WGSEF and its proximal operator.

**Weaknesses:**

- The paper's presentation and writing quality require improvement; the experimental section ends abruptly, seems the paper may be incomplete.

- The paper uses benchmark datasets like CIFAR10 and Fashion-MNIST, and benchmark architectures like VGG16, ResNet18, and MobileNetV1, which are standard choices for evaluating the performance of deep learning methods. However, these benchmarks might not fully reveal the practical efficiency and applicability of the proposed WGSEF regularization method in real-world, large-scale problems. For example, [Chen et al. 2021] tests its HSPG in structural prune ResNet on ImageNet, which could provide a stronger testament to the method's practicality.

- The process for grouping parameters before regularization is not specified. An exploration of automated grouping algorithms, such as in

Chen, Tianyi, et al. "OTOv2: Automatic, Generic, User-Friendly." The Eleventh International Conference on Learning Representations. 2022.

could be beneficial when applied with WGSEF.

**Questions:**

see the above weaknesses part

---

> ### Author Response · Authors · 2023-11-23
> **Reply to reviewer FGZh**
>
> We are pleased to address the points raised by reviewer FGZh.
>
> 1.	We have re-edited the related work, sections 5 and 6.
> 2.	We have added additional results for cifar100.
> 3.	An advantage of our framework over others is that it doesn’t require us/the user to “discover” groups. Groups naturally arise from the NN architecture, where in CNNs these are the filters, in a dense layer these are the columns, and in a transformer can be the attention heads. We are focused on the hardware, and the network weights can be represented as sparse matrices (e.g. SciPy BSR, CSC, CSR matrices), thus reducing the required disk space and RAM, and allowing faster inference. Since during inference, a block that represents a neuron is all zeros, there is no need to multiply and sum any incoming elements. Chen, Tianyi, et al. "OTOv2: Automatic, Generic, User-Friendly.", on the other hand, is a method that discovers Zero-Invariant Groups (ZIGs), which are groups that can be removed from the network while keeping the inter and intra-dimension “correct”, which is far less flexible than our method. OTO considers shrinking the network, we consider compression via sparsification, but we still consider advances in GPU hardware that allow speed-ups using sparse matrices. We gain flexibility, without compromising on the accuracy. Please see announcements from Nvidia in cuSPARSE:
>
> a.              Basic Linear Algebra for Sparse Matrices on NVIDIA GPUs
>
> b.              Accelerating Matrix Multiplication with Block Sparse Format and NVIDIA Tensor Cores

---

### Official Review · Reviewer_xL1V · 2023-11-03

**Soundness:** 2 fair
**Presentation:** 2 fair
**Contribution:** 2 fair
**Rating:** 3
**Confidence:** 3

**Summary:**

This paper proposes a weight regularization in deep learning with the goal of improving the hardware efficiency of neural networks. Specifically, they propose a method to learn structured and unstructured sparsity pattern in weights of neural network. They base their work on the sparse envelope function and extend it to the group case.

**Strengths:**

The paper provide details formulation and derivation of their method. Their experimental results show improvement against baselines based on Group Lasso to increasing group sparsity ratio.

**Weaknesses:**

- This work extend the sparse envelope function (Back & Refael (2022)) into group sparsity. So, it appears as an extension which is applied to deep learning training. To evaluate their theoretical contribution could the authors comment on how their analysis differs from the SEF? Perhaps, comment on the challenges in the extension from SEF to the group case. For example, some corollary are direct extension of derivations already done by SEF paper.

- What is the performance of WGSEF (algorithm 2) alone. This comparison is needed with WGSEF + HSPG to determine the advantages of proposed WGSEF.

- To evaluate the results, experiment are needed on popular deep learning architectures (Unet, Transformers). Please include standard deviation over the reported numbers. LeNet is a very basic network. Please update results with a neural network commonly used in the literature. Moreover, characterization of the method is missing. A figure on how the performance varies as a function of the hyper-parameter on the sparsity. This would be helpful for sensitivity analysis of the method.

- Almost all deep learning training are done with variants of Adam optimizer. I wonder if the authors could provide analysis on Adam as opposed to SGD.

- Literature review is not comprehensive. For example, literature on l1 convex relaxation to l0 is missing (one example [1]). It is not clear how the contribution of this work differs from prior work on inducing group sparsity for efficient neural network architecture.

- The organization of the paper needs improvement. Coherency can be improved. Here are some points. For example, in Section 5, the authors start talking about SGD which was not the main focus on the paper till this Section. Or again, the end of Section 5, the paper discusses some related works that are not relevant to the proposed method. I found many statements of the paper not related to the main theme of the paper, and also re-statement of results from prior works (e.g., Corollary 5.0.1). Moreover, I recommend to shorten the findings of Beck & Refael (2022) in the intro. Not clear why all those details are included in the intro before main contribution. Around (1) and (2), please mention that l0-norm is a pseudo norm. In find algorithm 1 redundant (Comparing to 2). Labels of figures are very small. Please increase the fontsize, and also use the table formatting suggested by the conference. I do find (1) and (2) unnecessary in the intro as the paper is based on SEF. Please justify why these equations must be there? I recommend to remove.

Other comments.

- Could the authors explain how the group SEF (their method) differs from group lasso? I find them related; however, the formulation no where refers to lasso or l1 type regularization.

- I thank the authors for their thorough related works; however, this introduction is missing a statement on "how their method differs from prior works". Please comment.

- Second paragraph of intro on generalization. Please also include advantages of over-parameterization. Otherwise, the sentence read a biased view. overparameterization does not result in an overfit (there is a double descent) if the amount of training data is very large.

- Please elaborate what the  "complete inverse problem" is (in the intro).

- Section 5 has related works on SGD which breaks down the flow of the paper. Could the authors explain why such related works on SGD is provided there? Recommend to move the paragraph after Algorithm 2 into appendix.

- Which experiment (dataset) results of Table 1 refer to.


[1] Donoho, D. L., & Elad, M. (2003). Optimally sparse representation in general (nonorthogonal) dictionaries via l1 minimization. Proceedings of the National Academy of Sciences, 100(5), 2197-2202.

**Questions:**

see above.

---

> ### Author Response · Authors · 2023-11-23
> **Reply to reviewer xL1V**
>
> We are pleased to address the points raised by reviewer xL1V.
> 1. First, we needed to find a way to extend SEF to a group structure. After-the-fact, imposing the L2 norm on the groups followed by the L0 norm on these groups norms may look natural, but this wasn’t the case when we started. We have considered different norms, and squared norms (which we realized creates a degenerated SEF instance). After reaching the final setup of the L2 norm on the groups, we still needed to develop an optimization algorithm. Adding group weights was not simple and for that reason, we began with fixed-size groups, however, since this was limiting we extended it again to allow arbitrary group sizes.
> 2. The results for the method solely were presented in each of the tables except for the first one, where the purpose of using HSPG was to make a similar comparison in terms of the optimization algorithm that incorporates the regularize.
> 3. We have added additional results for cifar100. We viewed ResNet / VGG / MobileNet as the main architectures, which in particular are being used in applications on edge devices where the HW resources are scarce. As for transformers, our method is very suitable for it, specifically for the attention blocks, and we would add these experiments in future works. However, the literature on NN sparsity specifically in structured sparsity is focused more on CNNs (and previously on RNNs). As we wanted to benchmark our work to SOTA structured sparsity algorithms, we focused on CNNs. Moreover, transformers are relatively new architecture, and only recently (~2 years) have reached their massive sizes of billions of parameters, so sparsity is getting more attention there only now, and thus there is less prior work to compare with.
> 4.
> a.	We have considered this option (variants of Adam optimizer), but unfortunately, our projection algorithm is not currently suitable to be used with Adam. Proximal-type algorithms solve the problem: $\min_v \{f(v) + \frac{1}{2}\|v - x\|\}$. With “standard” proximal methods such as proximal gradient or SGD-prox, we solve with regards to the L-2 norm, and our proof heavily relies on properties of the L-2 norm. In order to use Adam, we will need to solve the prox problem with regards to some D-norm, with D being a diagonali8 PD matrix. For details on a “prox-Adam” algorithm, please see Yun et al. “Adaptive proximal gradient methods for structured neural networks.”, section 2.
> b.	It’s important to note that we are still able to run the algorithm is less epochs than other benchmark methods, and achieve SOTA results. While it would be nice to use Adam, the optimization procedure itself is a means to an end, not the end itself.
> 5.
> a.	Thanks for your comment we extended the survey of the previous works in the revised version.
> b.	Our main contributions are stated in the abstract and intro and are as follows: We suggest an efficient algorithm to impose group sparsity, which is flexible to be used with any NN architecture. We allow to define groups by hardware properties so that sparsification yields significant inference speed-ups while keeping accuracy as-is. Moreover, we consider the simplicity of the algorithm as an advantage. Experimental results are good compared to SOTA.
> 6. We believe that a good sparsification algorithm / framework needs to also consider the optimization algorithm, otherwise it might be useless (for example, L0 norm which is intractable). Finding the right algorithm wasn’t as straightforward as anticipated, therefore we provide details on this subject.
> 7.	Group Lasso imposes L2 regularization on each group separately, without tying groups together. Therefore it imposes inter-group weight sharing, but not intra-group regularization. Our method, on the other hand, provides both types of regularizations in a holistic manner. Moreover, we believe the L0 formulation pushes the entire group values to be set to zero where this is the underlying real structure, while keeping positive groups almost unchanged (this is an intuition and not formally analyzed or proved).
> 8. The whole inverse problem simply means the whole optimization problem (that is basically an inverse problem - finding model parameters).
> 9. Thanks to the reviewer for the correct comment. Most deep learning applications on edge devices can be considered as in the interpolation threshold regime (high amount of parameters but not huge + the models were trained on a sufficient amount of training set), over-parameterization results in redundant variables without improving network results / accuracy. It requires more disk-space to store weights, more vRAM to load weights, and results in more FLOPs during inference, requiring more expensive hardware and more energy consumption, as well as environmental implications. This happens in main real-world deep learning applications, especially in end devices.
> 10. See bullet (6a) on this.
> 11. The datasets used are written inside the table - table 1 column 2.